# FPAW from *Trachinotus ovatus* Attenuates Potassium-Oxonate-Induced Hyperuricemia in Mice via Xanthine Oxidase Inhibition and Gut Microbiota Modulation: Molecular Insights and In Vivo Efficacy

**DOI:** 10.3390/nu17111831

**Published:** 2025-05-28

**Authors:** Huan Xiang, Dongxiao Sun-Waterhouse, Xiao Hu, Mengfan Hou, Shengjun Chen, Yanyan Wu, Yongqiang Zhao, Yueqi Wang

**Affiliations:** 1Key Laboratory of Aquatic Product Processing, Ministry of Agriculture and Rural Affairs, National R&D Center for Aquatic Product Processing, South China Sea Fisheries Research Institute, Chinese Academy of Fishery Sciences, Guangzhou 510300, China; skyxianghuan@163.com (H.X.); dxsun72@hotmail.com (D.S.-W.); houmengfan97@163.com (M.H.); chenshengjun@scsfri.ac.cn (S.C.); wuyygd@163.com (Y.W.); zhaoyq@scsfri.ac.cn (Y.Z.); wangyueqi@scsfri.ac.cn (Y.W.); 2Key Laboratory of Efficient Utilization and Processing of Marine Fishery Resources of Hainan Province, Sanya Tropical Fisheries Research Institute, Sanya 572018, China; 3School of Chemical Sciences, The University of Auckland, Private Bag 92019, Auckland 1010, New Zealand

**Keywords:** hyperuricemia, XOD inhibitory, gut microbiota, short-chain fatty acids, molecular docking

## Abstract

**Background:** Hyperuricemia (HUA) is a widespread metabolic disorder that arises from disruptions in purine metabolism, impaired kidney function, or both conditions. FPAW (Phe-Pro-Ala-Trp) is a novel peptide identified from *Trachinotus ovatus* with great XOD (xanthine oxidase) inhibitory activity (IC_50_ = 3.81 mM), which can be developed as a potential active ingredient to relieve hyperuricemia. However, it remains unclear whether FPAW alleviates HUA in vivo or not. **Methods:** In this study, potassium-oxonate-induced hyperuricemic mice were used to evaluate the in vivo anti-hyperuricemic activity of FPAW. Some physiological parameters, such as serum uric acid (SUA), serum creatinine (SCR), blood urea nitrogen (BUN), and the activity of XOD and ADA (adenosine deaminase) in the liver were determined to evaluate the effect of reduced uric acid. The modulations in the gut microbiota and its metabolites (SCFAs) were analyzed by sequencing the V3-V4 region of the 16S rRNA gene and GC-MS in different fecal samples. Molecular docking was used to predict the interactions between the enzymes and FPAW. **Results:** The results showed that FPAW reduced the levels of serum uric acid, serum creatinine, and blood urea nitrogen, while also suppressing the activity of XOD in the livers of HUA mice. Moreover, the FPAW treatment alleviated gut microbiota dysfunction and increased the production of short-chain fatty acids to protect normal intestinal function and health of the host. Molecular docking simulations revealed that FPAW inhibited XOD activity by entering the hydrophobic channel and interacting with amino acid residues on the surface via hydrogen bonding and hydrophobic interactions. **Conclusions:** This study provides new candidates for the development of hypouricemic drugs. FPAW exhibited great potential to relieve hyperuricemia of mice induced by diet in the animal experiment.

## 1. Introduction

Hyperuricemia is a disorder of purine metabolism characterized by excessive blood uric acid concentrations, which are generally brought on by either increased uric acid synthesis in the body or inadequate uric acid excretion. It is characterized by either overproduction or underexcretion of uric acid and is mainly manifested by long-term abnormal serum uric acid levels (>7.0 mg/dL (men) and 6.4 mg/dL (women)) [1,2]. Hyperuricemia is currently recognized as the fourth most significant chronic disease risk factor and has emerged as the second most prevalent metabolic disorder in China [3]. The current treatments for hyperuricemia are categorized into two types, one of which includes medications that decrease the production of uric acid, known as xanthine oxidase inhibitors, including allopurinol and febuxostat [4]. The other approach is to promote the excretion of uric acid using drugs such as benzosulfamulone, benbromarone, and some new urate transporter 1 inhibitors [5]. Nevertheless, these medications can cause a range of negative side effects, and prolonged use might result in liver and kidney damage, cardiovascular and cerebrovascular disorders, hypersensitivity syndrome, and other severe conditions that could threaten the patient’s life [6]. Therefore, there is an urgent need to develop natural product-derived alternatives for treating hyperuricemia.

Xanthine oxidase (XOD) belongs to the molybdenum hydroxylase flavoprotein family and is prevalent in mammalian tissues, particularly the liver and small intestine [7,8]. It is a key enzyme in uric acid production, facilitating the conversion of hypoxanthine to xanthine and ultimately to uric acid [4]. Using X-ray crystal diffraction, the crystal structure of XOD and its interactions with ligands have been elucidated. XOD is a homodimer with a molecular mass of approximately 300 kDa, consisting of two identical subunits [9]. Each subunit contains three catalytic centers: one molybdopterin (Mo-pt), two iron-sulfur centers (Fe-S), and flavin adenine dinucleotide (FAD) [10]. Xanthine oxidase is a primary focus for investigating the pathological processes and pharmacological effects related to hyperuricemia and gout [11] and is also popular today. XOD inhibitors can act on the purine-binding site [12] or FAD cofactor site [13], and the former is dominant and is a key site for purine oxidation and uric acid. In recent decades, short bioactive peptides have been shown to be easily absorbed, have significant beneficial effects, and can be used to regulate metabolic syndromes, such as hypertension [14]. Short peptides usually have high activity, stability, and specificity, are easy to mass produce, and are economical [15]. Therefore, bioactive peptides may serve as potential drugs for hyperuricemia treatment. Many bioactive peptides with the ability to lower uric levels have been identified in aquatic products such as *Sea cucumbers* [16] and tuna [17]. Bioactive peptides are considered to be natural XOD inhibitors with the advantages of high activity, high bio-affinity, and easy absorption [18]. Although a few XOD inhibitory peptides from natural sources have been identified, the action mechanisms of these peptides are not clear and need to be studied in depth.

The gut microbiota comprises the entire set of microorganisms that live in the intestine. In recent years, they have been considered important factors contributing to the development of hyperuricemia. Emerging research increasingly links gut microbiota to the pathogenesis of various diseases [19]. Numerous studies indicate that the diversity and abundance of gut microbiota change during the onset and alleviation of hyperuricemia. Uric acid can be excreted through the intestine; therefore, intestinal metabolites may be altered. Moreover, gut microbiota, such as *Lactobacillus* and *Pseudomonas*, are involved in purine and uric acid metabolism [20]. Previous research identified a significant increase in the relative abundance of *Firmicutes* and *Actinobacteria*, alongside a notable decrease in *Bacteroidetes* and *Proteobacteria*, in hyperuricemic mice [16]. Han et al. found that tuna meat oligopeptides reversed these changes, restoring *Firmicutes* and *Proteobacteria* while reducing *Bacteroidetes* and *Actinobacteria* [21]. Therefore, there must be a close link between hyperuricemia and gut microbiota.

In the previous study, we confirmed the potential anti-hyperuricemic properties of hydrolysates of *Trachinotus ovatus*. FAPW (Phe-Ala-Pro-Trp), a short bioactive peptide identified in *Trachinotus ovatus* hydrolysates, has been proven to be an effective inhibitor of XOD, with an IC_50_ of 3.81 ± 0.18 mM in a previous study [22], while the mechanism was not clear. Therefore, the molecular insights and in vivo efficacy of FPAW were evaluated via xanthine oxidase inhibition and gut microbiota modulation in this study. Potassium-oxonate-induced hyperuricemic mice were used to evaluate the in vivo anti-hyperuricemic activity of FPAW. Some physiological parameters, such as serum uric acid (SUA), serum creatinine (SCR), blood urea nitrogen (BUN), and the activity of XOD and ADA (adenosine deaminase) in the liver were determined to evaluate the effects of reduced uric acid. The modulations in the gut microbiota and its metabolites (SCFAs) were analyzed by sequencing the V3-V4 region of the16S rRNA gene and GC-MS (Gas Chromatography-Mass Spectrometry) in different fecal samples. Molecular docking was used to predict the interactions between the enzymes and FPAW.

## 2. Materials and Methods

### 2.1. Materials and Chemicals

FPAW, identified in *Trachinotus ovatus*, was synthesized using a solid-phase method by GL Biochemistry Co., Ltd. (Shanghai, China). The potassium oxonate and allopurinol were purchased from Shanghai Macklin Biochemical Technology Co., Ltd. (Shanghai, China). The isoflurane was purchased from Shenzhen Rayward Life Technology Co., Ltd. (Shenzhen, China).

### 2.2. Animal Experiment Design

Thirty-two male KM mice, each six weeks old and weighing 22 ± 1 g, were obtained from the Guangdong Medical Experimental Animal Center in Foshan, Guangdong, China (permit number: SCXK (Yue) 2023-0002). They were kept in a controlled setting at 24 °C with a 12 h light-dark cycle from 8:00 am to 8:00 pm. The mice were allowed free access to ultra-purified water and normal diet (Jiangsu Xietong Pharmaceutical Bio-Engineering Co., Ltd., Jiangsu, China). After a week of getting used to their surroundings, they were randomly assigned to one of four groups, each containing eight mice: control, model, positive control (allopurinol), and FPAW. The control group of mice was given saline orally once a day based on their weight, while the other groups received a potassium oxonate solution at a dosage of 250 mg/kg. Sixty minutes later, the mice in both the control and model groups received a specific amount of saline (0.9%, 80 mL/kg/d) via intragastric administration, while those in the allopurinol and FPAW groups were given allopurinol at a dose of 25 mg/kg and FPAW at 150 mg/kg, respectively, through oral gavage [23]. The experiment spanned 28 days, during which mice body weights were recorded bi-daily. After four weeks of feeding, fecal samples were collected in cryogenic vials and promptly stored at −80 °C for subsequent analysis. The mice were anesthetized with isoflurane, and blood samples were obtained via orbital puncture. The mice were then euthanized by cervical dislocation, and the liver and kidneys were excised, weighed, and frozen at −80 °C for later analysis. Liver and kidney indices were calculated based on the previous research [24].

### 2.3. Analysis of Biochemical Parameters

Serum samples were collected, after briefly being kept at room temperature, by centrifuging at 5000× *g* for 15 min at 4 °C and were subsequently stored at −80 °C for later physiological examination. The concentrations of uric acid, creatinine, and urea nitrogen were assessed according to Almeida et al. [25] by using detection kits (Nanjing Jiancheng Bioengineering Institute, Nanjing, China) following the manufacturer’s guidelines. The activities of XOD and ADA in the liver were measured according to the instructions provided by the same institute. Suzhou Keming Biotechnology Co., Ltd. (Suzhou, China) and Beijing League of Biotechnology Co., Ltd. (Beijing, China) were also involved in the process.

### 2.4. Total DNA Extraction, PCR, and Sequencing of Fecal Samples

The TIANcombi DNA Lyse&Det PCR Kit, produced by Tiangen Biotech (Beijing) Co., Ltd. in Beijing, China, was employed to extract total DNA from fecal samples according to Zhu et al. [26]. To amplify the V3-V4 regions of the bacterial 16S rRNA gene, primers B341F (5′-CCTACGGGNGGCWGCAG-3′) and B785R (5′-GACTACHVGGGTATCTAATCC-3′) were utilized. The resulting PCR products were then purified with the AGENCOURT^®^AMPURE^®^ XP Kit from Beckman Coulter, Inc., located in California, USA. Sequencing was conducted using an Illumina MiSeq system (Illumina, San Diego, CA, USA) at Shanghai Fuda Testing Technology Group Co., Ltd. (Shanghai, China).

The initial data underwent splicing, quality checks, and filtering before being compiled into operational taxonomic units (OTUs) with a 97% similarity threshold using the V-search software (v2.3.4). Sequences exhibiting more than 97% similarity were grouped into the same OTUs. Subsequently, the taxonomic annotation of the OUTs was completed after selecting representative sequences for comparison with the database. Species information with the highest similarity to OTU sequences and greater than 80% confidence was used for OTU annotation.

### 2.5. Measurement of the SCFA Content in Fecal Samples

Following a 12 h freeze-drying process, 50 mg of dried fecal samples were combined with 2 mL of a 3.68 mol/L phosphoric acid solution and mixed thoroughly using a vortex for 2 min. Next, 1 mL of ether was introduced to the reaction mixture and allowed to extract for 10 min. The mixture was then centrifuged at 3750× *g* for 20 min at a temperature of 4 °C, and the supernatant was collected. This extraction process was repeated once more, and the two resulting extraction solutions were combined. The solution was reduced to a volume of less than 1 mL by blowing nitrogen over it before being injected for analysis. SCFA levels were determined using GC-MS (ISQ LT, Thermo Fisher Scientific, MA, USA) equipped with a TG WAX (30 m × 0.25 mm × 0.25 μm) column. Helium served as the carrier gas at a flow rate of 1 mL/min with a split ratio of 75:1. The settings were as follows: the electron impact source voltage was 70 eV, single ion monitoring mode was employed, quantitative ions were 60 and 73, and the ion source temperature was maintained at 200 °C. A 10 mM volatile free acid mix, including acetic acid (C2), propionic acid (C3), butyric acid (C4), isobutyric acid (C4), isovaleric acid (C5), valeric acid (C5), isocaproic acid (C6, 4-methylvaleric acid), and caproic acid (C6), as well as 2-methylbutyric acid were purchased from Sigma-Aldrich (St. Louis, MO, USA), which were used as the standard SCFAs.

### 2.6. Molecular Docking Analysis

Molecular docking of XOD and ADA has been described in references [17,27], respectively. X-ray crystal structures of XOD with quercetin (PDB ID: 3NVY, 2.0 Å) and ADA with FR2 (PDB ID: 1NDW, 2.0 Å) were obtained from the RCSB Protein Data Bank. In XOD, chains D, E, F, and water molecules were removed using PyMOL, while non-standard residues and water molecules were excluded from ADA. Molecular docking was conducted using MGL tools 1.5.6 and AutoDock Vina. Polar hydrogens were added, and Gasteiger charges were assigned to protein atoms with ADT, setting the Zn atom charge in ADA to +2. For XOD, a 40 Å × 40 Å × 40 Å grid box was defined at coordinates x = 39.063, y = 21.898, and z = 20.218, with a grid spacing of 0.375 Å. ADA’s docking site was set at x = 49.0388, y = 55.4311, and z = 20.959. The docking outcomes were evaluated based on minimum energy. Ligand–enzyme interactions were visualized using Discovery Studio 2020.

### 2.7. Statistical Analysis

All experiments were performed in triplicate. The data processing was analyzed using one-way ANOVA, followed by Tukey’s post hoc test for multiple comparisons with SPSS 22.0 (International Business Machines Corporation, New York, NY, USA) and is shown as the mean ± standard deviation (SD). The correlations among the top 14 microbial genera and six SCFAs were analyzed by Pearson’s correlation tests using SPSS. Statistical significance was set at *p* < 0.05. The Venn diagram was drawn via http://bioinformatics.psb.ugent.be/webtools/Venn/ (accessed on 20 June 2024). The clustering heatmap was drawn by https://www.metaboanalyst.ca/faces/home.xhtml (accessed on 20 July 2024).

## 3. Results and Discussion

### 3.1. Effect of FPAW on Physiological Indexes of Hyperuricemic Mice

The body weight of mice reflects their physical condition and is an important indicator of whether an organism is healthy [28]. To explore the anti-hyperuricemic effects of FPAW, HUA rats were orally administered FPAW for 28 d. Throughout the experimental period, the mice remained in good health, and there were no notable differences in body weight among the groups (Figure 1A). In different stages of toxicity tests, the increases in animal body weight and organ weight were different, but there was a positive correlation between them. Therefore, the organ/body weight ratio can reflect the toxicity of drugs and the degree of influence on related organs to some extent, and it can also overcome the insufficiency of body weight changes. Figure 1B,C display the liver and kidney indices of the mice. In comparison to the control group, the liver index in the model control group showed a significant increase (*p* < 0.05), while no notable difference was found between the allopurinol and FPAW groups (*p* > 0.05). Additionally, there were no significant differences in kidney indices across the groups. These findings suggest that the treatment did not harm the liver or kidneys of the mice.

### 3.2. Effect of FPAW on Biochemical Indexes of Hyperuricemic Mice

The serum uric acid levels reflect the balance between uric acid production and excretion [29]. When uric acid cannot be excreted in a timely manner, it continues to accumulate in the body, resulting in a continuous increase in serum uric acid levels and hyperuricemia [30]. As illustrated in Figure 2A, the serum uric acid (SUA) level in the model group (99.12 ± 3.54 μmol/L) was notably higher than that of the control group (84.96 ± 3.54 μmol/L), with a significant difference (*p* < 0.05). Allopurinol was effective in significantly lowering the potassium-oxonate-induced SUA level to 68.44 ± 4.09 μmol/L, demonstrating a strong therapeutic impact on hyperuricemia. Furthermore, FPAW also decreased serum uric acid levels to 90.86 ± 2.04 μmol/L, with no significant difference observed compared to the control group (*p* > 0.05).

Serum creatinine (SCR) and BUN are waste products produced by the metabolism of muscle and protein, respectively, and are excreted by the kidneys. SCR and BUN levels are used to evaluate renal function in clinical settings, and their elevated levels are considered indicative of kidney impairment [31,32]. Figure 2B,C illustrate the SCR and BUN levels, respectively. In hyperuricemic mice, these levels were markedly elevated compared to those in healthy mice. Allopurinol treatment led to a significant reduction in SCR and BUN levels by 33.78% (9.69 ± 2.11 μmol/L) and 20.60% (0.48 ± 0.17 mg/mL), respectively, when compared to the model group (*p* < 0.05). In the FPAW group, the reductions were 22.53% (11.34 ± 1.17 μmol/L) and 12.68% (0.53 ± 0.24 mg/mL), respectively. These findings indicate that FPAW lowered SUA, SCR, and BUN levels, which may facilitate normal uric acid excretion, support the kidney’s normal physiological functions, and provide renal protection.

XOD and ADA are key enzymes involved in purine metabolism. The inhibition of their activity can directly affect uric acid formation. Allopurinol is the first-line drug for treating hyperuricemia. It is a structural isomer of hypoxanthine and an effective XOD inhibitor [33]. Figure 2D,E illustrate the activities of XOD and ADA. FPAW notably decreased the activities of XOD and ADA in liver tissue when compared to the model group, showing no significant difference from allopurinol. These findings suggest that FPAW serves as an effective enzyme inhibitor for XOD and ADA, thereby hindering purine metabolism and lowering uric acid production. Consequently, it represents a promising therapeutic target for mitigating this condition, which was consistent with the former report [34].

### 3.3. Effect of FPAW on the Modulation of the Gut Microbiota in Hyperuricemic Mice

The kidneys excrete two-thirds of the total uric acid produced in the body daily, whereas the remaining one-third is broken down by the gut microbiota and excreted in the stool [35]. Therefore, changes in the gut microbiota are important for lowering uric acid levels. We compared the gut microbial communities of mice in the experimental and control groups by sequencing the V3-V4 regions of the 16S rRNA gene. The OTU cluster analysis identified 2043 OTUs in the control group, 1640 in the model group, 1947 in the allopurinol group, and 2061 in the FPAW group. Figure 3A presents the Venn diagram. In comparison to the control group, the model group exhibited a notable reduction in species richness and diversity. However, both allopurinol and FPAW counteracted this decline, bringing the OTU count back to normal levels.

It has been widely reported that bioactive peptides can optimize the structure of intestinal flora and improve human health, mainly showing the increase of beneficial bacteria and the decrease of pathogenic bacteria [36]. Variations in gut microbiota structures were noted across different groups, as illustrated in Figure 3B,C. At the phylum level, *Bacteroidetes* and *Firmicutes* were the predominant phyla across all groups, with no notable differences observed between them. In comparison to the model group, the relative abundances of *Candidatus*, *Saccharibacteria*, and *Proteobacteria* significantly increased following FPAW treatment, while the other phyla did not show significant changes. The reduction of *Bacteroides* content has been confirmed to play a role in the recovery of HUA.

At the genus level, an analysis was conducted on 14 genera with higher abundance. The model group exhibited a higher relative abundance of *Muribaculaceae-unclassified*, *Lactobacillus*, and *Lachnospiraceae-unclassified bacteria* compared to the control group, whereas *Duncaniella* and *Bacteroidales-unclassified bacteria* were less abundant. In the FPAW group, the relative abundances of *Muribaculaceae-unclassified*, *Lactobacillus*, and *Duncaniella* were restored. In present study, FPAW treatment altered the overall structure of the gut microbiota in mice, suggesting that microflora mapping could help elucidate the mechanisms of FPAW against HUA, which is consistent with the results of a previous study showing that peptide changed the gut microbiota composition of mice with HUA [21].

### 3.4. Effect of FPAW on the Concentrations of SCFAs in Hyperuricemic Mice

Short-chain fatty acids (SCFAs) are primarily produced when the gut microbiota ferments non-digestible carbohydrates. In the human large intestine, the concentration of SCFAs typically ranges from 50 to 200 mmol/L, and they play significant roles in maintaining human health [37]. These include maintaining intestinal health [38], balancing metabolism, and regulating immunity [39]. Figure 4 illustrates the levels of six short-chain fatty acids (SCFAs): acetic, propionic, butyric, isobutyric, valeric, and isovaleric acids. The patterns of variation for all SCFAs were uniform across the different groups. In comparison to the control group, the concentrations of these SCFAs were markedly higher in the other groups, with the FPAW group showing the most significant increase. These findings suggest that feeding with FPAW has a notable impact on the metabolites produced by gut microbiota. While among the six SCFAs, acetic, propionic, butyric and valeric acids possess many health benefits such as anti-inflammatory, immunoregulatory, anti-obesity, anti-diabetes, anticancer, cardiovascular protective, hepatoprotective, and neuroprotective activities [40]. FPAW lowered SUA, SCR, and BUN levels, which may facilitate normal uric acid excretion, support the kidney’s normal physiological functions, and provide renal protection. FPAW exhibited great potential for relieving hyperuricemia in mice induced by diet in the animal experiment, which owed to the contribution of microbial community.

To illustrate the contribution of the microbial community to SCFAs during hyperuricemia treatment, the correlations among the top 14 microbial genera and six SCFAs were analyzed using Pearson’s correlation tests (Figure 5). For microbial genera, a cluster including *Lachnospiraceae-unclassified*, *Duncaniella*, *Mailhella*, *Ruminococcaceae-unclassified*, *Candidatus Saccharibacteria unclassified*, and Alistipes was markedly correlated with the concentration of SCFAs, whereas *Prevotellamassilia*, *Lactobacillus*, *Muribaculaceae-unclassified*, and *Ligilactobacillus* were significantly negatively correlated with SCFAs. As reported in previous studies, *Clostridiales* and *Ruminococcaceae* produce SCFAs [41]. The study found that the relative abundance of certain gut microbiota increased following FPAW treatment, aligning with the SCFA concentration. The findings indicated that FPAW could enhance the presence of specific gut microbiota and the levels of their metabolites (SCFAs).

### 3.5. Molecular Modeling Studies

Molecular modeling and docking analyses were employed to uncover how the ligand binds within the enzyme’s active site, with the findings depicted in Figure 6. Table 1 details all interactions between allopurinol/FPAW and XOD/ADA. In this research, FPAW was positioned into the hydrophobic cavity center of XOD, involving the following residues (Figure 6B). The affinity between them was −5.8 kcal/mol, indicating that they can combine well. According to the docking results, the interaction types of the XOD-FPAW complex were hydrogen bonding and hydrophobic. As shown in Figure 6B, FPAW was bonded to Ser876 of XOD via hydrogen bonding, and the atomic distance was 2.75 Å. A carbon–hydrogen bond was found between FPAW and Glu802, with a distance of 3.40 Å. The inhibitor was pushed into the hydrophobic channel mainly through hydrophobic interactions. Several types of hydrophobic interactions exist between them. The formation of π-π stacking with Phe914 is considered necessary for XOD activity because Phe914 residues located next to the molybdopterin center could also interact with the five-membered ring of xanthine [42]. In this study, two π–π stacking interactions were observed between the Trp residue in FPAW and Phe914 or Phe1009 of XOD, with distances of 4.40 Å and 4.73 Å, respectively. This result was similar to that obtained with allopurinol (Figure 6A).

Similarly, the interactions between the ligands and ADA were illustrated using molecular docking. Hydrogen bonding and hydrophobic interactions were the main interactions between ADA and FPAW, or allopurinol. Allopurinol formed two hydrogen bonds (interacting with Asp296 and Gly184) and π-π stacking with His17 of ADA (Figure 6C), whereas FPAW had a hydrogen bond and carbon hydrogen bond with Gly184 and a hydrogen bond with Glu217 (Figure 6D). There were also some hydrophobic bonds between them, such as π-sigma and π-alkyl bonds. However, there is no direct evidence that allopurinol and FPAW can combine with ADA to alter its spatial structure. FPAW may inhibit ADA activity by inhibiting XOD activity and then affecting purine metabolism.

## 4. Conclusions

FPAW demonstrates significant potential in mitigating diet-induced hyperuricemia in animal models. FPAW was found to effectively decrease the serum uric acid, serum creatinine and blood urea nitrogen level as well as inhibit the activity of xanthine oxidase adenosine deaminase of liver in hyperuricemic mice. Moreover, treatment with FPAW could also alleviate the dysfunction of the gut microbiota and increase the production of short-chain fatty acids to protect normal intestinal function and health. Molecular docking simulation showed that FPAW could inhibit XOD activity via entering into the hydrophobic channel and interacting with amino acid residues on the surface by hydrogen bond and hydrophobic effect. Consequently, FPAW holds promise as a novel and effective dietary source for uric-acid-lowering therapies. While the glomerular filtration rate and mechanism of lowering uric acid by FPAW is not mentioned in this study. The mechanism of key enzymes of the UA metabolic pathway in the liver of mice under the intervention of FPAW with the most significant uric acid lowering effect need further investigation.

## Figures and Tables

**Figure 1 nutrients-17-01831-f001:**
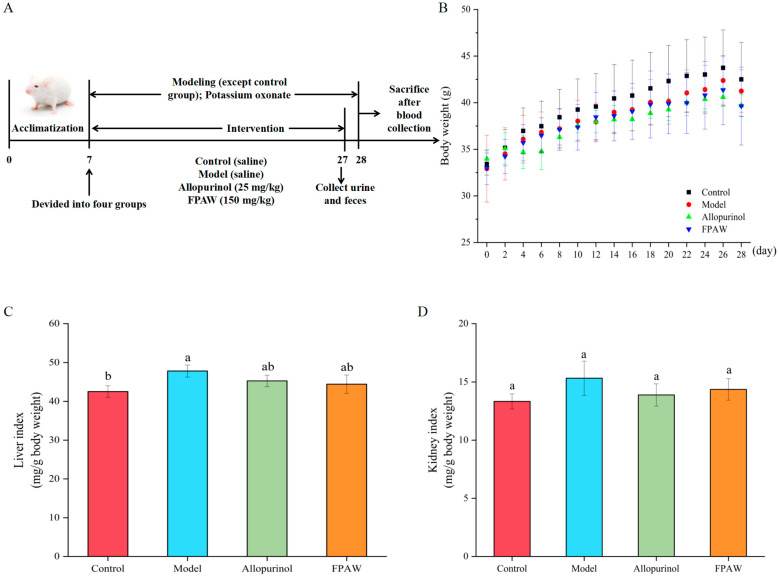
Effects of FPAW (Phe-Pro-Ala-Trp) on biochemical indexes and renal pathology of hyperuricemic mice. Experimental scheme of animals (**A**); body weight (**B**); liver index (**C**); and kidney index (**D**). The different letters in the figure indicate significant difference (*p* < 0.05).

**Figure 2 nutrients-17-01831-f002:**
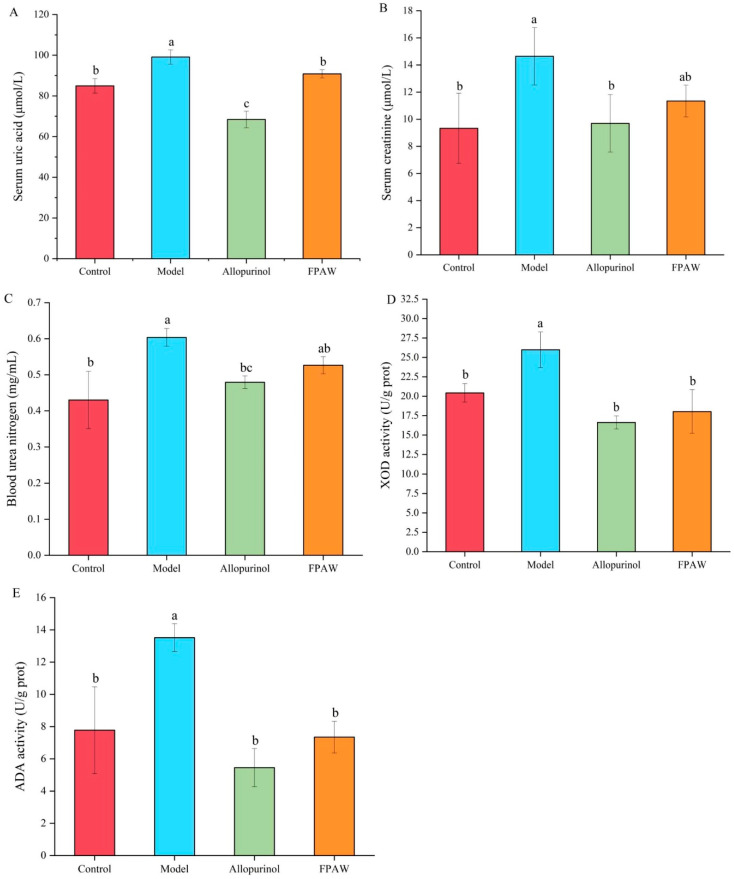
Effects of FPAW on the levels of serum uric acid (**A**), serum creatinine (**B**), and blood urea nitrogen (**C**) and the activity of xanthine oxidase (XOD) (**D**) and adenosine deaminase (ADA) (**E**) in hyperuricemic mice. The different letters in the figure indicate significant difference (*p* < 0.05).

**Figure 3 nutrients-17-01831-f003:**
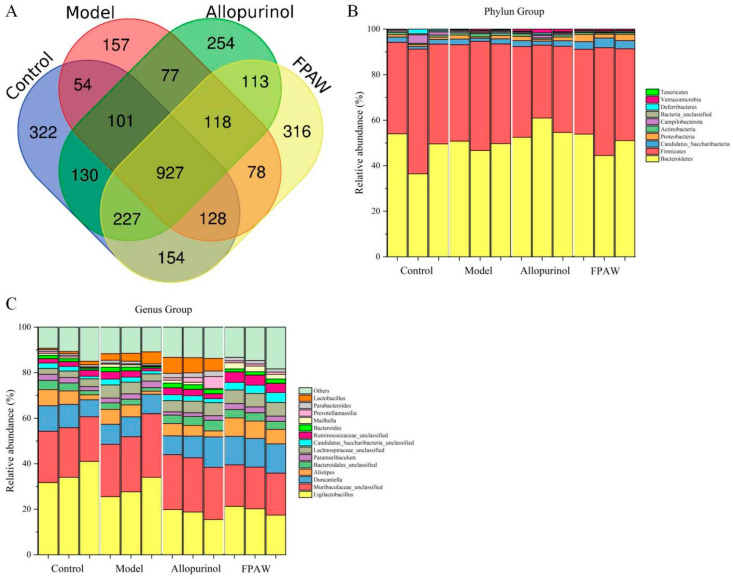
Effects of FPAW on the modulation of the gut microbiota structure in hyperuricemic mice. Venn diagram based on the OTU abundance of the microbial community (**A**), phylum level (**B**), and genus level (**C**).

**Figure 4 nutrients-17-01831-f004:**
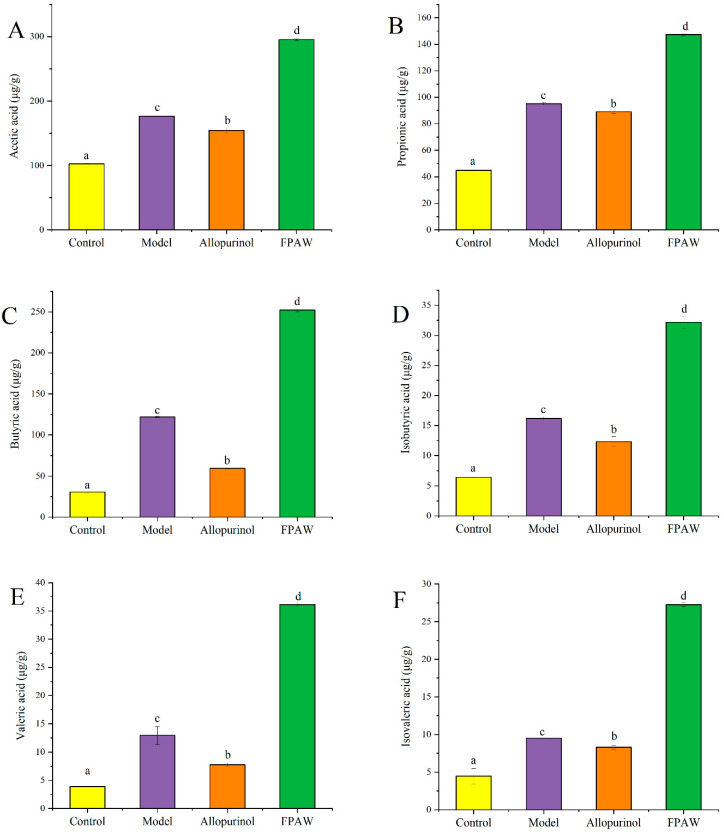
Effect of FPAW on the concentrations of SCFAs in hyperuricemic mice. (**A**) Acetic acid; (**B**) propionic acid; (**C**) Butyric acid; (**D**) Isobutyric acid; (**E**) Valeric acid; (**F**) Isovaleric acid. The different letters in the figure indicate significant difference (*p* < 0.05).

**Figure 5 nutrients-17-01831-f005:**
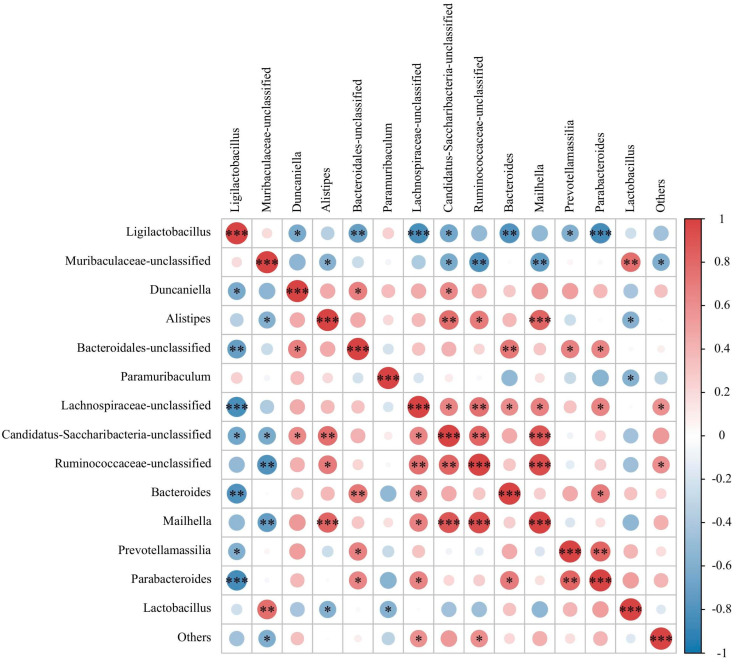
Clustering heatmap based on Pearson’s correlation tests among the top 14 microbial genera and six SCFAs. Darker red and blue indicate higher positive and negative correlations, respectively.

**Figure 6 nutrients-17-01831-f006:**
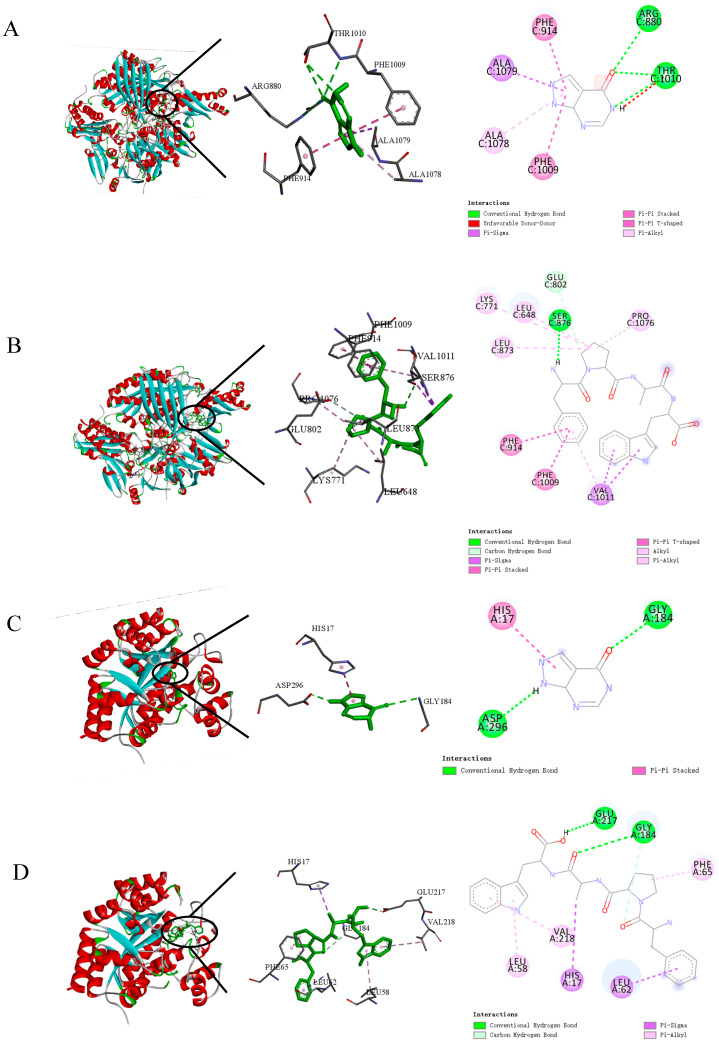
Molecular docking model and visualization analysis of allopurinol with xanthine oxidase (XOD) (**A**), FPAW with xanthine oxidase (XOD) (**B**), allopurinol with adenosine deaminase (ADA) (**C**), and FPAW with adenosine deaminase (ADA) (**D**). The pellets represent different amino acid residues of XOD, and different colors represent different interaction types.

**Table 1 nutrients-17-01831-t001:** Molecular docking of FPAW peptide and allopurinol with XOD and ADA protein explaining the intermolecular interactions and binding affinities, respectively.

Enzyme	Ligands	Energy (Kcal/mol)	Hydrogen Bond		Hydrophobic		
			Classical	Non-Classical	Pi-Hydrophobic	Alkyl-Hydrophobic	Mixed Hydrophobic
XOD	allopurinol	−6.5	Arg880 (3.22, 2.90, 3.15 Å); Thr1010 (3.19, 3.09 Å)		Phe1009 (4.70 Å); Phe914 (3.25 Å)		Ala1078 (4.83 Å) Ala1079 (3.80 Å)
	FPAW	−5.8	Ser876 (2.75 Å)	Glu802 (3.40 Å)	Phe1009 (4.73 Å); Phe914 (4.40 Å)	Pro1076 (4.42 Å) Lys771 (5.43 Å); Leu648 (5.49 Å) Leu873 (5.11 Å)	Val1011 (5.43, 3.74, 3.74, 3.89 Å)
ADA	allopurinol	−5.2	Asp296 (2.08 Å); Gly184 (2.96 Å)		His17 (3.57 Å)		
	FPAW	−9.2	Gly184 (3.10 Å); Glu217 (2.07 Å)	Gly184 (3.64 Å)			Val218 (4.94 Å); Leu58 (5.06 Å); His17 (3.63 Å); Phe65 (3.83 Å); Leu62 (3.92 Å)

## Data Availability

Data are contained within the article.

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
