# Peer review of "FPAW from Trachinotus ovatus Attenuates Potassium-Oxonate-Induced Hyperuricemia in Mice via Xanthine Oxidase Inhibition and Gut Microbiota Modulation: Molecular Insights and In Vivo Efficacy"

_nutrients, 2025, doi:10.3390/nu17111831_

Round 1

Reviewer 1 Report

Comments and Suggestions for Authors

In this work, potassium oxonate-induced hyperuricemic mice were used to evaluate the in vivo anti-hyperuricemic activity of FPAW, a short bioactive peptide identified from Trachinotus ovatus hydrolysates. After reading this work thoroughly, I have some major observations:

  1. Line 80: Avoid using abbreviations only at the first mention.
  2. Line 102: what is meant by “a specific amount of saline”? Could you be more specific?
  3. Line 133: OUT or OTU?
  4. Figure captions: Avoid using abbreviations when possible.
  5.  Conclusion: Rewrite with focus on the main findings for clarity.
  6. There are some typos and grammatical mistakes throughout the manuscript. It would be beneficial to proofread the text.
Comments on the Quality of English Language

There are some typos and grammatical mistakes throughout the manuscript. It would be beneficial to proofread the text.

Author Response

Detailed Responses to the Editor and Reviewers

Comments 1 Line 80: Avoid using abbreviations only at the first mention.

Response 1: Agree. thanks for your suggestion, we have added the full name in line 80, FAPW (Phe-Ala-Pro-Trp), a short bioactive peptide identified from Trachinotus ovatus hydrolysates.

Comments 2 Line 102: what is meant by “a specific amount of saline”? Could you be more specific?

Response 2: Agree. thanks for your question, we have added the specific amount of saline Sixty minutes later, the mice in both the control and model groups received a specific amount of saline (0.9%, 80 mL/kg/d) via intragastric administration, while those in the allopurinol and FPAW groups were given allopurinol at a dose of 25 mg/kg and FPAW at 150 mg/kg, respectively, through oral gavage.

Comments 3 Line 133: OUT or OTU?

Response 3: Agree. thanks for your question, we have checked it and ensured it was OTUs. Initial data underwent splicing, quality checks, and filtering before being compiled into operational taxonomic units (OTUs) with a 97% similarity threshold using the V-search software.

Comments 4 Figure captions: Avoid using abbreviations when possible.

Response 4: Agree. thanks four your suggestion, we have checked the abbreviations and added the full name in the figure captions.

Figure 1. Effects of FPAW (Phe-Pro-Ala-Trp) on biochemical indexes and renal pathology of hyperuricemic mice. Experimental scheme of animals (A); Body weight (B); Liver index (C) and Kidney index (D). 

Figure 2. Effects of FPAW on the levels of serum uric acid (A), serum creatinine (B), and blood urea nitrogen (C), and the activity of Xanthine oxidase (XOD) (D) and adenosine deaminase (ADA) (E) in hyperuricemic mice. 

Figure 6. Molecular docking model and visualization analysis of allopurinol with Xanthine oxidase (XOD) (A), FPAW with Xanthine oxidase (XOD) (B), allopurinol with adenosine deaminase (ADA) (C), and FPAW with adenosine deaminase (ADA) (D). The pellets represent different amino acid residues of XOD, and different colors represent different interaction types.

Comments 5: Conclusion: Rewrite with focus on the main findings for clarity.

Response 5: Agree. thanks for your suggestion. We have revised the conclusion part as follows: FPAW demonstrates significant potential in mitigating diet-induced hyperuricemia in animal models. FPAW was found to effectively decrease the serum uric acid, serum creatinine and blood urea nitrogen level as well as inhibit the activity of xanthine oxidase adenosine deaminase of liver in hyperuricemic mice. Moreover, treatment with FPAW could also alleviate the dysfunction of the gut microbiota and increase the production of short chain fatty acid to protect normal intestinal function and health. Molecular docking simulation showed that FPAW could inhibit XOD activity via entering into the hydrophobic channel and interacting with amino acid residues on the surface by hydrogen bond and hydrophobic effect. Consequently, FPAW holds promise as a novel and effective dietary source for uric acid-lowering therapies.

Comments 6: There are some typos and grammatical mistakes throughout the manuscript. It would be beneficial to proofread the text.

Response 6: Agree. thanks for your suggestion, we have checked the manuscript and revised with red words. Such as in Line 15, line 23-26, line 34-36, line 38-39 et al.

Molecular docking simulations revealed that FPAW inhibited XOD activity by entering the hydrophobic channel and interacting with amino acid residues on the surface via hydrogen bonding and hydrophobic interactions. This study provides new candidates for the development of hypouricemic drug.

Hyperuricemia is currently recognized as the fourth most significant chronic disease risk factor and has emerged as the second most prevalent metabolic disorder in China [3].

The other approach is to promote the excretion of uric acid using drugs such as benzosulfamulone, benbromarone, and some new urate transporter 1 inhibitors.

Reviewer 2 Report

Comments and Suggestions for Authors

The authors evaluated the effects of the FPAW peptide on hyperuricemia in a mouse model. The manuscript concludes that FPAW may ameliorate hyperuricemia by inhibiting xanthine oxidase (XOD) activity and modulating gut microbiota composition. While the study addresses a relevant topic, several major and minor issues must be addressed to improve the manuscript's scientific clarity and structural integrity.

Major Concerns:

  1. Title: The hyperuricemia model used in this study was induced by potassium oxonate, not diet-induced hyperuricemia. Please revise the title accordingly to reflect the actual model used.
  2. Abstract Structure: The abstract lacks a clear Methods section, which disrupts the logical flow. Please include a brief description of the methodology used.
  3. Abbreviation Usage: Abbreviations such as FPAW and XOD must be fully defined upon first mention in the abstract and the main text. Please carefully check and revise throughout the manuscript.
  4. Introduction Structure: The Introduction section lacks clarity and logical organization. Please restructure this section to clearly outline the background, rationale, and objectives of the study.
  5. FPAW Dosage Justification: Please provide a scientific rationale or citation for the selected dosage of FPAW used in this study.
  6. Figure 1A – Experimental Design: The experimental timeline shown in Figure 1A does not match the description in the text. Please clarify the exact duration of FPAW administration. Was it administered for only two weeks?
  7. Data Interpretation – Renal Markers: The manuscript claims a significant reduction in serum creatinine and BUN levels by FPAW (lines 19–20 and 208–210), but the data show no statistical significance. Please revise this claim to reflect the results accurately.
  8. Lack of In-depth Discussion: The Discussion section lacks depth. While the authors restate the findings, there is little interpretation or contextual analysis. Please expand this section to discuss the implications and potential mechanisms more thoroughly.
  9. SCFA Analysis: Six SCFAs were significantly increased by FPAW treatment, even exceeding levels in the positive control group. Since not all increased SCFAs benefit human health, this should be addressed and discussed critically in the manuscript.

Minor Concerns:

  1. Line 40 – Missing Citation: Please provide a proper reference to support the statement.
  2. Units – Line 204 (and elsewhere): Please revise the unit expression from "rpm" to "g" (relative centrifugal force) for centrifugation speed, as this is the standard unit in biomedical research.

Author Response

The authors evaluated the effects of the FPAW peptide on hyperuricemia in a mouse model. The manuscript concludes that FPAW may ameliorate hyperuricemia by inhibiting xanthine oxidase (XOD) activity and modulating gut microbiota composition. While the study addresses a relevant topic, several major and minor issues must be addressed to improve the manuscript's scientific clarity and structural integrity.

Comments 1: Title: The hyperuricemia model used in this study was induced by potassium oxonate, not diet-induced hyperuricemia. Please revise the title accordingly to reflect the actual model used.

Responses 1: Agree. The hyperuricemia model used in this study was induced by potassium oxonate, not diet-induced hyperuricemia. And we have changed the title to “FPAW from Trachinotus ovatus Attenuates Potassium Oxonate-Induced Hyperuricemia in Mice via Xanthine Oxidase In-hibition and Gut Microbiota Modulation: Molecular Insights and In Vivo Efficacy”

Comments 2: Abstract Structure: The abstract lacks a clear Methods section, which disrupts the logical flow. Please include a brief description of the methodology used.

Response 2: Agree. Thanks for your suggestion, we have added the methodology we used in the abstract. In this study, potassium oxonate-induced hyperuricemic mice were used to evaluate the in vivo anti-hyperuricemic activity of FPAW. Some physiological parameters, such as serum uric acid (SUA), serum creatinine (SCR), blood urea nitrogen (BUN), and the activity of XOD and ADA in the liver, were determined to evaluate the effect of re-duced uric acid. The modulations in the gut microbiota and its metabolites (SCFAs) were analyzed by sequencing the V3-V4 region of the16S rRNA gene and GC-MS in different fecal samples. Molecular docking was used to predict the interactions be-tween the enzymes and the FPAW.

Comments 3: Abbreviation Usage: Abbreviations such as FPAW and XOD must be fully defined upon first mention in the abstract and the main text. Please carefully check and revise throughout the manuscript.

Response 3: Agree. Thanks for your suggestion, we have checked and added the full name in the all manuscript. FPAW (Phe-Pro-Ala-Trp) was a novel peptide identified from Trachinotus ovatus with great XOD (Xanthine oxidase) inhibitory activity (IC50 = 3.81 mM) which can be developed as a potential active ingredient to relieve hyperuricemia.

In this study, potassium oxonate-induced hyperuricemic mice were used to evaluate the in vivo anti-hyperuricemic activity of FPAW. Some physiological parameters, such as serum uric acid (SUA), serum creatinine (SCR), blood urea nitrogen (BUN), and the activity of XOD and ADA (adenosine deaminase) in the liver, were determined to evaluate the effect of re-duced uric acid. The modulations in the gut microbiota and its metabolites (SCFAs) were analyzed by sequencing the V3-V4 region of the16S rRNA gene and GC-MS in different fecal samples. Molecular docking was used to predict the interactions be-tween the enzymes and the FPAW.  

Comments 4: Introduction Structure: The Introduction section lacks clarity and logical organization. Please restructure this section to clearly outline the background, rationale, and objectives of the study.

Response 4: Agree. Thanks for your suggestion, we have restructured this section. Hyperuricemia is caused by purine metabolism disorders, kidney dysfunction, or both disorders. It is characterized by either overproduction or underexcretion of uric acid and is mainly manifested by long-term abnormal serum uric acid levels (> 7.0 mg/dL (men) and 6.4 mg/dL (women)) [1,2]. Hyperuricemia is currently recognized as the fourth most significant chronic disease risk factor and has emerged as the second most prevalent metabolic disorder in China [3]. Current treatments for hyperuricemia are categorized into two types: medications that decrease the production of uric acid, known as xanthine oxidase inhibitors, including allopurinol and febuxostat [4]. The other approach is to promote the excretion of uric acid using drugs such as benzosulfamulone, benbromarone, and some new urate transporter 1 inhibitors. Nevertheless, these medications can cause a range of negative side effects, and prolonged use might result in liver and kidney damage, cardiovascular and cerebrovascular disorders, hypersensitivity syndrome, and other severe conditions that could threaten the patient’s life.[6]. Therefore, there is an urgent need to develop natural product-derived alternatives for treating hyperuricemia.

Xanthine oxidase (XOD) belongs to the molybdenum hydroxylase flavoprotein family and is prevalent in mammalian tissues, particularly the liver and small intestine [7]. It is a key enzyme in uric acid production, which mainly catalyzes the conversion of hypoxanthine to xanthine and the latter to uric acid [4]. Using X-ray crystal diffraction, the crystal structure of XOD and its interactions with ligands have been elucidated. XOD is a homodimer with a molecular mass of approximately 300 kDa, consisting of two identical subunits [8]. Each subunit contains three catalytic centers: one molybdopterin (Mo-pt), two iron-sulfur centers (Fe-S), and flavin adenine dinucleotide (FAD) [9]. Xanthine oxidase is a primary focus for investigating the pathological processes and pharmacological effects related to hyperuricemia and gout [10], and is also popular today. XOD inhibitors can act on the purine-binding site [11] or FAD cofactor site [12], and the former is dominant and is a key site for purine oxidation and uric acid. In recent decades, short bioactive peptides have been shown to be easily absorbed, have significant beneficial effects, and can be used to regulate metabolic syndromes, such as hypertension [13]. Short peptides usually have high activity, stability, and specificity, are easy to mass produce, and are economical [14]. Therefore, bioactive peptides may serve as potential drugs for hyperuricemia treatment. Many bioactive peptides with the ability to lower uric levels have been identified in aquatic products such as sea cucumbers [15] and tuna [16]. Bioactive peptides are considered to be natural XOD inhibitors with the advantages of high activity, high bio-affinity and easy absorption [17]. Although a few XOD inhibitory peptides from natural sources have been identified, the action mechanism of these peptides are not clear and need to be studied in depth.

The gut microbiota comprises the entire set of microorganisms that live in the intestine. In recent years, they have been considered important factors contributing to the development of hyperuricemia. Emerging research increasingly links gut microbiota to the pathogenesis of various diseases [18]. Numerous studies indicate that the diversity and abundance of gut microbiota change during the onset and alleviation of hyperuricemia. Uric acid can be excreted through the intestine; therefore, intestinal metabolites may be altered. Moreover, gut microbiota, such as Lactobacillus and Pseudomonas, are involved in purine and uric acid metabolism [19]. Previous research identified a significant increase in the relative abundance of Firmicutes and Actinobacteria, alongside a notable decrease in Bacteroidetes and Proteobacteria, in hyperuricemic mice [15]. Han et al. found that tuna meat oligopeptides reversed these changes, restoring Firmicutes and Proteobacteria while reducing Bacteroidetes and Actinobacteria [20]. Therefore, there must be a close link between hyperuricemia and gut microbiota.

In the previous study, we have confirmed the potential anti-hyperurinemic properties of hydrolysates of Trachinotus ovatus, FAPW (Phe-Ala-Pro-Trp), a short bioactive peptide identified in Trachinotus ovatus hydrolysates, has been proven to be an effective inhibitor of XOD, with an IC50 of 3.81 ± 0.18 mM in a previous study [21] , while the mechanisim was not clearly. Therefore, the molecular insights and in vivo efficacy of FPAW were evaluated via xanthine oxidase in-hibition and gut microbiota modulation in this study. Potassium oxonate-induced hyperuricemic mice were used to evaluate the in vivo anti-hyperuricemic activity of FPAW. Some physiological parameters, such as serum uric acid (SUA), serum creatinine (SCR), blood urea nitrogen (BUN), and the activity of XOD and ADA in the liver, were determined to evaluate the effect of reduced uric acid. The modulations in the gut microbiota and its metabolites (SCFAs) were analyzed by sequencing the V3-V4 region of the16S rRNA gene and GC-MS in different fecal samples. Molecular docking was used to predict the interactions between the enzymes and the FPAW.

Comments 5: FPAW Dosage Justification: Please provide a scientific rationale or citation for the selected dosage of FPAW used in this study.

Response 5: Agree. Thanks for your suggestion, we have added the citation for the selected dosage of FPAW used in this study. Which was in line 116-119, Sixty minutes later, the mice in both the control and model groups received a specific amount of saline (0.9%, 80 mL/kg/d) via intragastric administration, while those in the allopurinol and FPAW groups were given allopurinol at a dose of 25 mg/kg and FPAW at 150 mg/kg, respectively, through oral gavage [22]. The reference was Zhou, J.; Wang, Z.; Zhang, Z.; Han, J.; Feng, Y.; Zhang, J.; Zhang, Z.; Li, Y.; Ming, T.; Lu, C.; et al. Modulation of Gut Microbiota and Serum Metabolome by Apostichopus Japonicus Derived Oligopeptide in High-Fructose Diet-Induced Hyperuricemia in Mice. Food Science and Human Wellness 2025, 14, 9250011, doi:10.26599/FSHW.2024.9250011.

Comments 6: Figure 1A – Experimental Design: The experimental timeline shown in Figure 1A does not match the description in the text. Please clarify the exact duration of FPAW administration. Was it administered for only two weeks?

Response 6: Agree. Thanks for your suggestion, we have checked the Figure 1 and the experimental time line, and we have fixed it as follows.

Comments 7: Data Interpretation – Renal Markers: The manuscript claims a significant reduction in serum creatinine and BUN levels by FPAW (lines 19–20 and 208–210), but the data show no statistical significance. Please revise this claim to reflect the results accurately.

Response 7: Agree. Thanks for your suggestion, we have checked the data and revise the claim in line 26-28: The results showed that FPAW reduced the levels of serum uric acid, serum creatinine, and blood urea nitrogen, while also suppressing the activity of XOD in the livers of HUA mice. Line 225-227, These findings indicate that FPAW lowered SUA, SCR, and BUN levels, which may facili-tate normal uric acid excretion, support the kidney's normal physiological functions, and provide renal protection. 

Comments 8: Lack of In-depth Discussion: The Discussion section lacks depth. While the authors restate the findings, there is little interpretation or contextual analysis. Please expand this section to discuss the implications and potential mechanisms more thoroughly.

Response 8: Agree. We have added the discussion more deeply.

In line 228-236, XOD and ADA are key enzymes involved in purine metabolism. Inhibition of their activity can directly affect uric acid formation. Allopurinol is the first-line drug for treating hyperuricemia. It is a structural isomer of hypoxanthine and an effective xanthine oxidase inhibitor [30]. Figure 2 D and Figure 2 E illustrate the activities of XOD and ADA. FPAW notably decreased the activities of XOD and ADA in liver tissues when compared to the model group, showing no significant difference from allopurinol. These findings suggest that FPAW serves as an effective enzyme inhibitor for XOD and ADA, thereby hindering purine metabolism and lowering uric acid production. Consequently, it represents a promising therapeutic target for mitigating this condition, which was in consisted with the former report [31].

In line 253-269, It has been widely reported that bioactive peptides can optimize the structure of intes-tinal flora and improve human health, mainly showing the increase of beneficial bacteria and the decrease of pathogenic bacteria [33]. Variations in gut microbiota structures were noted across different groups, as illustrated in Figure 3B and 3C. At the phylum level, Bac-teroidetes and Firmicutes were the predominant phyla across all groups, with no notable differences observed between them. In comparison to the model group, the relative abun-dances of Candidatus, Saccharibacteria, and Proteobacteria significantly increased following FPAW treatment, while the other phyla did not show significant changes. At the genus level, an analysis was conducted on 14 genera with higher abundance. The model group exhibited a higher relative abundance of Muribaculaceae-unclassified, Lactobacillus, and Lachnospiraceae-unclassified bacteria compared to the control group, whereas Duncaniella and Bacteroidales-unclassified bacteria were less abundant. In the FPAW group, the relative abundances of Muribaculaceae-unclassified, Lactobacillus, and Duncaniella were restored. In present study, FPAW treatment altered the overall structure of the gut microbiota in mice, suggesting that microflora mapping could help elucidate the mechanisms of FPAW against HUA, which is consistent with the results of a previous study that peptide changed the gut microbiota composition of mice with HUA [34].

In line 288-296, For microbial genera, a cluster including Lachnospiraceae-unclassified, Duncaniella, Mailhella, Ruminococcaceae-unclassified, Candidatus Saccharibacteria unclassified, and Alistipes was markedly correlated with the concentration of SCFAs, whereas Prevotellamassilia, Lactoba-cillus, Muribaculaceae-unclassified, and Ligilactobacillus were significantly negatively corre-lated with SCFAs. As reported in previous studies, Clostridiales and Ruminococcaceae pro-duce SCFAs [38]. The study found that the relative abundance of certain gut microbiota in-creased following FPAW treatment, aligning with the SCFA concentration. The findings indicated that FPAW could enhance the presence of specific gut microbiota and the levels of their metabolites (SCFAs).

Comments 9: SCFA Analysis: Six SCFAs were significantly increased by FPAW treatment, even exceeding levels in the positive control group. Since not all increased SCFAs benefit human health, this should be addressed and discussed critically in the manuscript.

Response 9: Agree. Thanks for your suggestion, we have checked this part and revised as follows. Figure 4 illustrates the levels of six short-chain fatty acids (SCFAs): acetic, propionic, butyric, isobutyric, valeric, and isovaleric acids. The patterns of variation for all SCFAs were uniform across the different groups. In comparison to the control group, the con-centrations of these SCFAs were markedly higher in the other groups, with the FPAW group showing the most significant increase. These findings suggest that feeding with FPAW has a notable impact on the metabolites produced by gut microbiota. While among the six SCFAs, acetic, propionic, butyric and valeric possess many health bene-fits such as anti-inflammatory, immunoregulatory, anti-obesity, anti-diabetes, anti-cancer, cardiovascular protective, hepatoprotective, and neuroprotective activities [39]. FPAW lowered SUA, SCR, and BUN levels, which may facilitate normal uric acid ex-cretion, support the kidney's normal physiological functions, and provide renal protec-tion. FPAW exhibited great potential to relieve hyperuricemia of mice induced by diet in the animal experiment, which was owing to the contribution of microbial commu-nity.   

Comments 10: Line 40 – Missing Citation: Please provide a proper reference to support the statement.

Response 10: Agree. Thanks for your suggestion, we have revise this part, and insert the reference. Hyperuricemia is caused by purine metabolism disorders, kidney dysfunction, or both disorders. It is characterized by either overproduction or underexcretion of uric acid and is mainly manifested by long-term abnormal serum uric acid levels (> 7.0 mg/dL (men) and 6.4 mg/dL (women)) [1,2]. Hyperuricemia is currently recognized as the fourth most significant chronic disease risk factor and has emerged as the second most prevalent metabolic disorder in China [3]. Current treatments for hyperuricemia are categorized into two types: medications that decrease the production of uric acid, known as xanthine oxidase inhibitors, including allopurinol and febuxostat [4]. The other approach is to promote the excretion of uric acid using drugs such as benzosulfamulone, benbromarone, and some new urate transporter 1 inhibitors [5].

Comments: Units – Line 204 (and elsewhere): Please revise the unit expression from "rpm" to "g" (relative centrifugal force) for centrifugation speed, as this is the standard unit in biomedical research.

Response 11: Agree. Thanks for your suggestion, we have changed the rpm to g in the manuscript.

Reviewer 3 Report

Comments and Suggestions for Authors

The comments follow throughout the attached document.

Author Response

Comment 1: Missing from the abstract of aim and methodology. Add the full name of the abbreviation.

Response 1: Agree. Thanks for your suggestion, we have added the aim and methodology in the abstract part, also the full name of abbreviations were included. Abstract: Hyperuricemia (HUA) is a widespread metabolic disorder that arises from disruptions in purine metabolism, impaired kidney function, or both conditions. FPAW (Phe-Pro-Ala-Trp) was a novel peptide identified from Trachinotus ovatus with great XOD (Xanthine oxidase) inhibitory activity (IC50 = 3.81 mM) which can be developed as a potential active ingredient to relieve hyperuricemia. However, it remains unclear whether FPAW alleviates HUA in vivo or not. In this study, potassium oxonate-induced hyperuricemic mice were used to evaluate the in vivo anti-hyperuricemic activity of FPAW. Some physiological parameters, such as serum uric acid (SUA), serum creatinine (SCR), blood urea nitrogen (BUN), and the activity of XOD and ADA (adenosine deaminase) in the liver, were determined to evaluate the effect of reduced uric acid. The modulations in the gut microbiota and its metabolites (SCFAs) were analyzed by sequencing the V3-V4 region of the16S rRNA gene and GC-MS in different fecal samples. Molecular docking was used to predict the interactions between the enzymes and FPAW. The results showed that FPAW reduced the levels of serum uric acid, serum creatinine, and blood urea nitrogen, while also suppressing the activity of XOD in the livers of HUA mice. Moreover, FPAW treatment alleviated gut microbiota dysfunction and increased the production of short-chain fatty acids to protect normal intestinal function and health of the host. Molecular docking simulations revealed that FPAW inhibited XOD activity by entering the hydrophobic channel and interacting with amino acid residues on the surface via hydrogen bonding and hydrophobic interactions. This study provides new candidates for the development of hypouricemic drug. FPAW exhibited great potential to relieve hyperuricemia of mice induced by diet in the animal experiment.

Comment 2 Hyperuricemia is caused by purine metabolism disorders, kidney dysfunction, or 31 both. Among other causes, it seems that there are only these two causes. It is more correct if you change the sentence.

Response 2: Agree. Thanks for your suggestion, we have revised this sentence as follows: Hyperuricemia is a disorder of purine metabolism characterized by excessive blood uric acid concentrations, which are generally brought on by either increased uric acid synthesis in the body or inadequate uric acid excretion.

Comment 3: It is a key enzyme in uric acid production, which mainly catalyzes the conversion of hy-53 poxanthine to xanthine and the latter to uric acid [4]. Please check and revise.

Response 3: Agree. Thanks for your suggestion, we have revised this sentence as follows: It is a key enzyme in uric acid production, facilitating the conversion of hypoxanthine to xanthine and ultimately to uric acid [4].

Comment 4: the last paragraph of introduction needs to revise.

Response 4: Agree. Thanks for your suggestion, we have revised this paragraph as follows: In the previous study, we have confirmed the potential anti-hyperurinemic properties of hydrolysates of Trachinotus ovatus, FAPW (Phe-Ala-Pro-Trp), a short bioactive peptide identified in Trachinotus ovatus hydrolysates, has been proven to be an effective inhibitor of XOD, with an IC50 of 3.81 ± 0.18 mM in a previous study [22], while the mechanisim was not clearly. Therefore, the molecular insights and in vivo efficacy of FPAW were evaluated via xanthine oxidase in-hibition and gut microbiota modulation in this study. Potassium oxonate-induced hyperuricemic mice were used to evaluate the in vivo anti-hyperuricemic activity of FPAW. Some physiological parameters, such as serum uric acid (SUA), serum creatinine (SCR), blood urea nitrogen (BUN), and the activity of XOD and ADA in the liver, were determined to evaluate the effect of reduced uric acid. The modulations in the gut mi-crobiota and its metabolites (SCFAs) were analyzed by sequencing the V3-V4 region of the16S rRNA gene and GC-MS (Gas Chromatography-Mass Spectrometry) in different fe-cal samples. Molecular docking was used to predict the interactions between the enzymes and the FPAW.

Comment 5: Phe-Pro-Ala-Trp (FPAW), This acronym has already been used.

Response 5: Agree. Thanks for your question, we have deleted the full name of FPAW.

Comment 6: You must indicate the ethical issues regarding the animal study developed.

Response 6: Agree. Thanks for your suggestion, we have uploaded the animal statement to the editor.

Comment 7: The mice were allowed unlimited access to food and water. Indicate the company, city and country that sold this food. For all reagents, products, equipment and software, indicate this way. Bottled water?

Response 7: Agree. Thanks for your suggestion, we have revised this sentence. The mice were allowed with free access to water and normal diet (Jiangsu Xietong Pharmaceutical Bio-Engineering Co., Ltd., Jiangsu, China).

Comment 8: Why was the glomerular filtration rate not assessed or estimated?

Response 8: Agree. We have checked and revised the section 2.3. For kidney function, blood urea nitrogen (BUN) and blood creatinine (Cr) are key indicators for assessing glomerular filtration rate and renal excretory function. Therefore, the glomerular filtration rate not assessed.

Comment 9: detection kits from Nanjing Jiancheng Bioengineering Institute, located in Nanjing, Jiangsu, China, following the manufacturer's guidelines. Manually, or did you use automated methods? indicate method references!

Response 9: Agree. The determine method was claimed in the kit, therefore we determined the parameters according to the kit is fine.

Comment 10: You must include references to the methods or procedures used. In section 2.4.

Response 10, Agree. We have added the reference in this part as follows: The TIANcombi DNA Lyse&Det PCR Kit, produced by Tiangen Biotech (Beijing) Co., Ltd in Beijing, China, was employed to extract total DNA from fecal samples ac-cording to Zhu et al [25].

Comment 11: section 2.5, Describe the parameters and metabolites analyzed.

Response 11: Agree. Thanks for your suggestion, we have added the missing message. 10 mM volatile free acid mix, including acetic acid (C2), propionic acid (C3), butyric acid (C4), isobutyric acid (C4), isovaleric acid (C5), valeric acid (C5), isocaproic acid (C6, 4-methylvaleric acid), and caproic acid (C6), and 2-methylbutyric acid were pur-chased from Sigma-Aldrich (St. Louis, MO, USA), which were used as the standard SCFA.

Comment 12: All experiments were performed in triplicate. Data processing was analyzed using one-way ANOVA with SPSS 22.0 and is shown as the mean ± standard deviation (SD). What statistical test was used, the Tukey test? Indicate the company, city and country.

Response 12: Agree. Thanks for your suggestion, we have fixed as follows. All experiments were performed in triplicate. Data processing was analyzed using one-way ANOVA (Turkey) with SPSS 22.0 (International Business Machines Corporation, America) and is shown as the mean ± standard deviation (SD).

Comment 13: The results should be more discussed and compared with other studies.

Response 13: Agree, thanks for your suggestion, we have fixed this part as follows.

Serum creatinine (SCR) and BUN are waste products produced by the metabolism of muscle and protein, respectively, and are excreted by the kidneys. SCR and BUN levels are used to evaluate renal function in clinical settings, and their elevated levels are considered indicative of kidney impairment [30], [31]. Figure 2 B and Figure 2 C illustrate the SCR and BUN levels, respectively. In hyperuricemic mice, these levels were markedly elevated compared to those in healthy mice. Allopurinol treatment led to a significant reduction in SCR and BUN levels by 33.78% (9.69 ± 2.11 μmol/L) and 20.60% (0.48 ± 0.17 mg/mL), respectively, when compared to the model group (p < 0.05). In the FPAW group, the reductions were 22.53% (11.34 ± 1.17 μmol/L) and 12.68% (0.53 ± 0.24 mg/mL), respectively. These findings indicate that FPAW lowered SUA, SCR, and BUN levels, which may facilitate normal uric acid excretion, support the kidney's normal physiological functions, and provide renal protection.

XOD and ADA are key enzymes involved in purine metabolism. Inhibition of their activity can directly affect uric acid formation. Allopurinol is the first-line drug for treating hyperuricemia. It is a structural isomer of hypoxanthine and an effective xanthine oxidase inhibitor [32]. Figure 2 D and Figure 2 E illustrate the activities of XOD and ADA. FPAW notably decreased the activities of XOD and ADA in liver tissues when compared to the model group, showing no significant difference from allopurinol. These findings suggest that FPAW serves as an effective enzyme inhibitor for XOD and ADA, thereby hindering purine metabolism and lowering uric acid production. Consequently, it represents a promising therapeutic target for mitigating this condition, which was in consisted with the former report [33].

It has been widely reported that bioactive peptides can optimize the structure of intestinal flora and improve human health, mainly showing the increase of beneficial bacteria and the decrease of pathogenic bacteria [35]. Variations in gut microbiota structures were noted across different groups, as illustrated in Figure 3B and 3C. At the phylum level, Bacteroidetes and Firmicutes were the predominant phyla across all groups, with no notable differences observed between them. In comparison to the model group, the relative abundances of Candidatus, Saccharibacteria, and Proteobacteria significantly increased following FPAW treatment, while the other phyla did not show significant changes. The reduction of Bacteroides content has been confirmed to play a role in the recovery of HUA.

At the genus level, an analysis was conducted on 14 genera with higher abundance. The model group exhibited a higher relative abundance of Muribaculaceae-unclassified, Lactobacillus, and Lachnospiraceae-unclassified bacteria compared to the control group, whereas Duncaniella and Bacteroidales-unclassified bacteria were less abundant. In the FPAW group, the relative abundances of Muribaculaceae-unclassified, Lactobacillus, and Duncaniella were restored. In present study, FPAW treatment altered the overall structure of the gut microbiota in mice, suggesting that microflora mapping could help elucidate the mechanisms of FPAW against HUA, which is consistent with the results of a previous study that peptide changed the gut microbiota composition of mice with HUA [36].

Comment 14: figure captions, You must mention the meaning of the letters for all figures and results.

Response 14: Agree, thanks for your suggestion, we have added the meaning of the letters for all figures and results.

Figure 1. Effects of FPAW (Phe-Pro-Ala-Trp) on biochemical indexes and renal pathology of hyperuricemic mice. Experimental scheme of animals (A); Body weight (B); Liver index (C) and Kidney index (D). The different letter in the figure means the significant difference (p < 0.05)

Comment 15: in conclusion part, Also refer to the glomerular filtration rate and why was it not evaluated in this study? And in this sense, must indicate limitations of the study.

Response 15: Agree. Thanks for your question, we have checked the manuscript and find that for kidney function, blood urea nitrogen (BUN) and blood creatinine (Cr) are key indicators for assessing glomerular filtration rate and renal excretory function. Therefore, the glomerular filtration rate not assessed.

Round 2

Reviewer 2 Report

Comments and Suggestions for Authors

The authors had addressed the reviewer's concerns, and the manuscript can be accepted for publication.

Author Response

thanks for your positive comment.

Reviewer 3 Report

Comments and Suggestions for Authors

Dear all,

Most of the suggested changes were made, but some were not, so please proceed as suggested in the attached document.

Author Response

comment 1:  In this study, potassium oxonate-induced hyperuricemic mice were used to evaluate the in vivo anti-hyperuricemic activity of FPAW. put the in vivo to italics.

response 1: Agree. the sentence was fix to  In this study, potassium oxonate-induced hyperuricemic mice were used to evaluate the in vivo anti-hyperuricemic activity of FPAW. 

comment 2: Therefore, the molecular insights and in vivo efficacy of FPAW were evaluated via xanthine oxidase in-hibition and gut microbiota modulation in this study. put the in vivo to italics.

response 2: Agree. the sentence was fixed to Therefore, the molecular insights and in vivo efficacy of FPAW were evaluated via xanthine oxidase in-hibition and gut microbiota modulation in this study.

comment 3: Some physiological parameters, such as serum uric acid (SUA), serum creatinine (SCR), blood urea nitrogen (BUN), and the activity of XOD and ADA in the liver, were determined to evaluate the effect of reduced uric acid.  identity the ADA.

response 3: Agree, we have added the full name of ADA. Some physiological parameters, such as serum uric acid (SUA), serum creatinine (SCR), blood urea nitrogen (BUN), and the activity of XOD and ADA (Adenosine Deaminase) in the liver, were determined to evaluate the effect of reduced uric acid.

comment 4: The mice were allowed with free access to water and normal diet (Jiangsu Xietong Pharmaceutical Bio-Engineering Co.,Ltd., Jiangsu, China). Please indicate the type of water.

response 4: Agree. thanks for your suggestion, we have fixted to The mice were allowed with free access to ultra purified water and normal diet (Jiangsu Xietong Pharmaceutical Bio-Engineering Co.,Ltd., Jiangsu, China).

comment 5: line 133. Manually, or did you use automated methods? indicate method references.  The acronym ADA has already been used!

response 5: Agree. we have added the reserence and revised this sentence. The concentrations of uric acid, creatinine, and urea nitrogen were assessed according to Almeida et al [25] by using detection kits (Nanjing Jiancheng Bioengineering Institute, Nanjing, Jiangsu, China) following the manufacturer's guidelines. The activities of XOD and ADA in the liver were measured according to the instructions provided by the same institute. Suzhou Keming Biotechnology Co., Ltd. (Suzhou, Jiangsu, China) and Beijing League of Biotechnology Co., Ltd. (Beijing, China) were also involved in the process.

comment 6, the 4000 g was not convert.

response 6: Agree. we have fixted it in line 161. The mixture was then centrifuged at 3750 g for 20 minutes at a temperature of 4 °C, and the supernatant was collected.

comment 7: Put it this way: " ANOVA, followed by Tukey’s post hoc test for multiple comparisons." missing city, country.

response 7:agree. Data processing was analyzed using one-way ANOVA, followed by Tukey’s post hoc test for multiple comparisons with SPSS 22.0 (International Business Machines Corporation, New York, America) and is shown as the mean ± standard deviation (SD). 

comment 8: line 239, put XOD

response 8: Agree.  we have changed to XOD.

comment 9: put The different letter in the figure means the significant difference (p < 0.05). in figure 2, figure 4.

response 9: Agree, Figure 2. Effects of FPAW on the levels of serum uric acid (A), serum creatinine (B), and blood urea nitrogen (C), and the activity of Xanthine oxidase (XOD) (D) and adenosine deaminase (ADA) (E) in hyperuricemic mice. The different letter in the figure means the significant difference (p < 0.05).

Figure 4. Effect of FPAW on the concentrations of SCFAs in hyperuricemic mice. The different letter in the figure means the significant difference (p < 0.05).

comment 10: This correlation was not indicated in the methodology in line 305.

response 10: Agree. The correlations among the top 14 microbial genera and six SCFAs were analyzed using Pearson’s correlation tests by using SPSS. it was in line 191-192.

comment 11: Also refer to the glomerular filtration rate and why was it not evaluated in this study? And in this sense, must indicate limitations of the study.

response 11: For kidney function, blood urea nitrogen (BUN) and blood creatinine (Cr) are key indicators for assessing glomerular filtration rate and renal excretory function. Therefore, the glomerular filtration rate not assessed.

While the glomerular filtration rate and mechanism of lowering uric acid by FPAW is not mentioned in this study. The mechanism of key enzymes of UA metabolic pathway in liver of mice under the intervention of FPAWwith the most significant uric acid lowering effect need further investigation. 
